# Perforating Gout: Expanding the Differential for Transepidermal Elimination

**Michal Bohdanowicz and Scott H. Bradshaw ***

Department of Medicine, Sunnybrook Campus, University of Toronto, 2075 Bayview Ave., Toronto, ON M4N 3N5, Canada; m.bohdanowicz@mail.utoronto.ca
* Correspondence: scott.bradshaw@sunnybrook.ca; Tel.: +1-613-453-1860

**Abstract:** Perforating dermatoses are dermatologic disorders with transepidermal elimination (TE) of dermal substances. While TE is typically associated with collagen and elastin, it can also occur as a secondary event in other processes, and it is important to keep a broad differential. We present a case of perforating tophaceous gout, which underscores the need for a thoughtful approach to perforating disorders. An updated review of recent literature is also presented.

**Keywords:** gout; tophus; perforating dermatosis; transepidermal elimination; pseudoepitheliomatous hyperplasia; granulomatous dermatitis

## 1. Introduction

Perforating dermatoses are a group of dermatologic disorders characterized by transepidermal elimination (TE) of dermal substances, including collagen, elastin, and other connective tissue components [1]. Less commonly, perforation can be due to the elimination of other materials, including calcium, mucin, foreign bodies, and even cellular material such as melanocytes. Granulomatous inflammation can also lead to perforation, examples of which include granuloma annulare, sarcoid nodules, rheumatoid nodules, and granulomatous infections. The epidermal changes associated with TE can be a diagnostic pitfall, as superficial biopsies can mimic other entities such as squamous cell carcinoma.

## 2. Case Report

We present a biopsy from the knee of a 38-year-old man with a hyperkeratotic, non-healing lesion. On low power, there is marked pseudoepitheliomatous hyperplasia with a crust (Figure 1a). There is a central defect with downgrowth of the epidermis towards pale dermal deposits and TE of the deposits. On high power (Figure 1b), the deposits are composed of fluffy amphiphilic material, surrounded by a palisade of histiocytes. A diagnosis of perforating tophaceous gout was made.

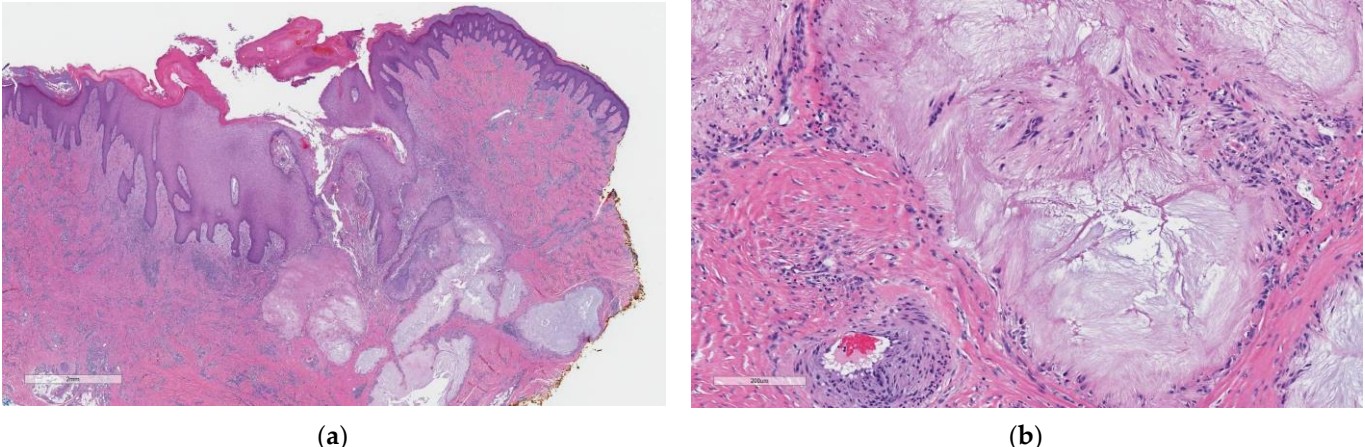

(**a**)                                                                                                              (**b**)

**Figure 1.** (**a**) Hematoxylin and eosin-stained skin at low magnification shows pseudoepithelioma-tous hyperplasia and degenerating material abutting the epidermal downgrowth. (**b**) At high power, fluffy amphiphilic material surrounded by a palisade of histiocytes is visible.

### 3. Discussion

*3.1. Approach to Perforating Dermatoses*

In this case, ample material was provided by the clinician, and although there was clinical uncertainty, the histologic diagnosis was straightforward. This underscores the value of a generous biopsy, especially when an unusual diagnosis is present.

Perforating dermatoses are a group of dermatologic disorders characterized by the TE of dermal substances, including collagen, elastin, and other connective tissue components. Woo et al. [1] point out that not all material eliminated from the dermis through the epidermis represents TE. TE is characterized by a remodeling of the epidermis to form downgrowths towards abnormal material within the dermis. The epidermis envelops the abnormal material, and subsequent maturation of keratinocytes transports the material from the dermis to the epidermis. Multiple channels are thus formed by the remodeled epidermis.

This process is contrasted against a suppurative ulcer, which is characterized by the destruction of the epidermis and subsequent elimination of dermal material through a draining sinus or ulcer. Material from the dermis is eliminated, but it is not transported by keratinocytes through TE.

The issue gets confused as perforating dermatoses often have central zones where the epidermis is absent (i.e., ulcerated). The resulting ulcer will develop a crust and a neu-trophilic inflammatory response to some degree. For TE, this neutrophilic inflammation should be a minor component rather than a draining abscess.

Returning to an approach to TE, the first step is to recognize the pseudoepithelioma-tous hyperplasia as potentially representing the epidermal remodeling of TE. This can be challenging if the specimen does not include the full thickness of the epidermis. In addi-tion, TE can mimic in-situ squamous cell carcinoma, both clinically, as in this case, and histologically, especially if the biopsy is superficial [2].

The second step in the approach is the examination of the eliminated substance. Again, a biopsy that transects the epidermis may not include the underlying substance. In our ex-perience, this is commonly the case for biopsies to evaluate chondrodermatitis nodularis helicis versus squamous cell carcinoma. Although the diagnosis can typically be made on superficial biopsies, a sample including the entire epidermis is preferable.

Sorting causes of TE into categories provides an organized approach:

1. Endogenous substances (e.g., collagen, elastin)
2. Abnormal deposits (e.g., calcium, mucin)
3. Granulomatous disorders (e.g., sarcoid)
4. Cellular elimination (e.g., nevus)

The "primary" perforating disorders involve the elimination of endogenous dermal material (category 1 above). These include elastosis perforans serpiginosis, reactive perforating collagenosis, perforating folliculitis, and Kyrle's disease. Although these conditions may share similar histologic features in terms of a mixed inflammatory infiltrate and hyperkeratosis, they differ in terms of their eliminated substance.

Elastosis perforans serpiginosa involves the TE of elastic fibers. The fibers may be subtle, and the diagnosis can be confirmed by identifying the presence of elastic fibers within the epidermis using special stains such as the Verhoeff-Van Gieson stain. Because elastosis perforans serpiginosis most often occurs on the lateral neck and flexural skin, the location of the biopsy is a helpful clue.

The perforating substance in reactive perforating collagenosis is collagen (Figure A1). In our experience, some patients exhibit extrusion of degenerating collagen and elastin into the crust overlying the ulcer in the early phase of wound healing after routine biopsies or excisions. It is therefore important to recognize that not all ulceration represents TE [1]. For a diagnosis of perforating collagenosis, there should be remodeling/invagination of the epidermis. Ideally, the extrusion of collagen and/or elastic fibers between keratinocytes in the intact epidermis adjacent to the ulcer can also be demonstrated, as shown in Figure A1.

The most common cause of perforating collagenosis is renal failure, but there are a number of other associations including diabetes, malignancy, rheumatoid arthritis, and other systemic diseases.

Kyrle's disease is characterized by the TE of keratin and other materials. As with perforating collagenosis and elastosis perforans, there is often a history of diabetes or renal disease. There is longstanding controversy regarding the classification of Kyrle's disease [2], specifically whether it represents a distinct entity or if it exists on a spectrum with the other TE disorders often associated with renal disease and/or diabetes [3]. We support the latter view, and our histologic reports use the umbrella term "acquired perforating dermatosis."

Perforating folliculitis shows a similar pattern of TE of elastin and/or collagen, but the elimination channels are specifically associated with the follicular epithelium.

The remaining categories 2–4 are considered secondary causes of TE or perforation. Examples of these conditions include perforating granuloma annulare (Figure A2), perforating calcinosis, and perforating amyloidosis, among others (Table 1). Some entities reported to give rise to TE may belong to more than one category. For example, granuloma annulare has an abnormal accumulation of mucin, but the granuloma formation is felt to be the more important process leading to TE, thus it has been placed in the "granulomatous" category in the table. A similar argument applies to necrobiosis lipoidica and ochronosis. Conversely, while most infectious agents associated with TE do so through granuloma formation, botryomycosis does not typically show a prominent granulomatous reaction, and TE is more likely related to the elimination of the abnormal granule, putting it in the "abnormal deposit" category rather than the "granulomatous" category. This organism is associated with suppurative inflammation, and some cases of perforation may actually represent simple draining abscesses and not true TE.

Perforating pilomatricomas are also difficult to classify. This tumor characteristically shows rupture with an associated granulomatous reaction, and therefore it may fit into the "granulomatous" category. Dystrophic calcification is also common, which would put it in the "abnormal deposit" category along with calcinosis cutis and calcified elastic tissue. In some cases, the elimination appears to represent abnormal cells. As with all the entities in the granulomatous category, some cases may not represent TE at all but, in fact, be simple ulceration.

**Table 1.** Categorization of perforating disorders based on the source of the TE material.

| **Primary** | | |
|---|---|---|
| **Source** | **Disease** | **TE Material** |
| Endogenous material | Elastosis perforans serpiginosa | Elastin |
| | Reactive perforating collagenosis | Collagen |
| | Kyrle's disease | Keratin and other material |
| | Perforating folliculitis | Elastin and/or Collagen |
| | Acquired perforating dermatosis | Umbrella term for the entities listed above |
| **Secondary** | | |
| **Source** | **Disease** | **TE Material** |
| Abnormal endogenous material | Chondrodermatitis nodularis helicis | Degraded cartilage |
| | Perforating pseudoxanthoma elasticum | Abnormal elastin |
| | Perforating osteoma cutis | Ectopic bone |
| | Perforating calcinosis cutis | Ectopic calcium |
| | Perforating calcific collagenosis | Calcified collagen |
| | Perforating calcific elastosis | Calcified elastin |
| Deposits | Perforating mucinosis | Mucin |
| | Perforating amyloidosis | Amyloid |
| | Perforating cryocrystalglobulinemia | Cryocrystalglobulins |
| Infection with deposits | Botryomycosis | Extra-bacterial granule |
| Granulomas | Perforating granuloma annulare | Necrobiotic granulomas with mucin |
| | Perforating necrobiosis lipoidica | necrobiotic granulomas |
| | Perforating sarcoidosis | Sarcoidal granulomas |
| | Perforating rheumatoid nodule | Granuloma, fibrinoid |
| | Perforating foreign body | Foreign body granulomas and foreign material |
| | Perforating tattoo reaction | Granulomatous inflammation with tattoo pigment |
| | Perforating ochronosis | Granulomatous inflammation with homogentisic acid |
| | Perforating gout | Granulomatous inflammation with monosodium urate |
| Granulomatous Infections | Schistosomiasis | Granulomatous inflammation |
| | Chromomycosis | Granulomatous inflammation |
| | Tuberculosis | Granulomas |
| | Leprosy | Granulomas |
| | Leishmaniasis | Granulomatous inflammation |
| | Histoplasmosis | Granulomatous inflammation |
| Other | Lichen nitidus | Inflammatory cells |
| | Melanoma | Melanocytes |
| | Nevus | Melanocytes |
| | Pilomatricoma | abnormal immature keratinocytes |

Several perforating diseases are caused by abnormal endogenous material. Calcified collagen or elastin can cause perforating calcific collagen or elastosis. Often, these are induced by the application of exogenous substances containing calcium.

Other perforating disorders are due to deposits of inorganic or organic material. The major inorganic material that perforates is calcium. Calcium can deposit in the dermis by different mechanisms, including dystrophic, metastatic, and idiopathic calcification. Perforating calcinosis cutis is likely on a spectrum with calcific collagenosis and elastosis depending on the degree of calcification. Many organic molecules can also deposit in the dermis and become targets for TE. Amyloid, a byproduct of misfolded proteins, is one such molecule. Most commonly, keratinocytes are the source of amyloid in perforating amyloidosis, and scratching is thought to be a major etiologic driver for these conditions. This deposition prefers colder areas of the body, like the skin. Homogentisic acid, on the other hand, accumulates in parts of the skin that have been exposed to chemicals like hydroquinone.

Exogenous material such as sutures, splinters, plant material, or glass is commonly perforated through the epidermis. In some cases, this may represent incidental drainage of an underlying abscess, but it may also induce a foreign body reaction that leads to granulomas and true TE, especially under conditions where the foreign material cannot be absorbed. Sarcoidosis is a systemic condition with idiopathic granulomas. Unlike necrobiosis lipoidica and granuloma annulare, where granulomas occur in association with other material, sarcoidosis has isolated granulomas. In some cases, these granulomas can penetrate the epidermis.

Another rare source of epidermal perforation is tumor cells. As previously mentioned, perforating pilomatricomas may fit into this category, along with melanocytes, with TE being reported in association with both nevi and melanoma. It is less clear what triggers these cells to leave the body via the epidermis. For example, whether they aberrantly express a migratory protein is a question left for speculation. Interestingly, pagetoid spread in a melanoma may reflect transepidermal elimination of melanoma cells and define a key characteristic of malignancy.

Gout is not typically included in itemized lists of causes for TE, but it has been rarely reported [4–7]. By using the categorical approach here, however, one can extrapolate that any granulomatous process could potentially be in the differential diagnosis. An organized categorical approach thus allows a broader differential to be considered.

In this case, the H&E histology was classic, and confirmatory testing was not required. If there were less material or less classic histology, the diagnosis could be confirmed using polarized light microscopy on a fresh smear or unstained section prepared with alcohol or Carnoy fixative. The crystals have negative birefringence, confirming that the material represents monosodium urate crystals. Performing a De Galantha stain, again using tissue fixed with alcohol or Carnoy fixative, is another method [7], but it is rarely used today.

*3.2. Review of Recent Literature*

We reviewed the literature over the past four years to provide an update on new developments in perforating dermatoses and TE. As discussed above, TE is not synonymous with ulceration, and perforation is sometimes applied loosely to cases of simple ulceration [8–10].

Most of the literature relates to primary perforating dermatosis. Two large series [11,12] confirm renal disease to be by far the most common cause of perforating dermatosis, followed by diabetes and malignancy. Many case reports and smaller series of acquired perforating dermatoses were identified [13–25], again most commonly associated with renal disease and/or diabetes [26–36]. Associations with other systemic diseases have also been observed. The most common disease was rheumatoid arthritis [37–39], but dermatomyositis [40] and lupus [41] were also reported.

Association with malignancy continues to be reported [42–45].

With the onset of immunotherapy and targeted therapy, several drugs have been implicated in reactive perforating dermatosis, including interferon [46], vemurafenib [47], erlotinib [48], and a PD-1 inhibitor [49]. Associations with immunobullous diseases have also been noted [50–52]. Less common associations include pregnancy [53,54], non-red tat-

toos [55], MRSA [56], copper deficiency [57], Down's syndrome [58], and rhabdomyolysis-related hypercalcemia [59]. One report noted acquired perforating dermatosis in a pair of siblings [60], suggesting that there may be a genetic component involved in the etiology. Many reports focus on the treatment of perforating dermatoses [61–73], but this is beyond the scope of this report.

There has been much less discussion of secondary causes of TE. The most common reports are of granuloma annulare [74–76] and other granulomatous processes, including necrobiosis lipoidica [77,78], sarcoid [79], and elimination of suture material [80]. Other reports include perforating pseudoxanthoma elasticum [81–83], pilomatrixoma [84–86], sialolithiasis [87,88], lichen nitidus [89,90], and osteoma cutis [91]. Unusual reports of perforating cryocrystalglobulinemia in the context of multiple myeloma [92] and perforating ochronosis [93] again underscore the importance of a categorical approach over rigid adherence to previously published lists. While primary perforating dermatoses typically occur in older patients, several reports of secondary perforating processes have been noted in children [94–96].

Finally, several reports note that secondary perforating dermatoses can be mistaken for other entities, including squamous cell carcinoma [4], psoriasis [97,98], a nevus [99], and soft tissue malignancy [100,101]. Necrotizing infundibular crystalline folliculitis [102–104] can mimic perforating gout, having a cup-shaped surface (due to a distended follicular infundibulum) containing needle-like crystals.

## 4. Summary

Most of the recent literature reiterates known types of perforating dermatoses and known associations. Primary reactive dermatoses, including elastosis perforans serpiginosis, perforating collagenosis, perforating folliculitis, and Kyrle's disease, are most commonly associated with renal insufficiency and diabetes. Less common associations include malignancy and systemic diseases such as rheumatoid arthritis. Immunotherapy and targeted cancer treatments have emerged as new causes of perforating dermatitis.

Our case of perforating gout represents a rare and unusual secondary perforating dermatosis. The diagnosis of perforating dermatoses can be diagnostically challenging, and diagnostic pitfalls include squamous cell carcinoma, psoriasis, and sarcoma.

Reliance on lists of causes of perforating dermatoses is sometimes inadequate, and a categorical approach is better able to accommodate rare causes of TE, such as gout. This case report also emphasizes the value of a generous biopsy. In our case, the gouty tophus is only present at a depth of 3 mm, which would be missed entirely by most shave biopsies.

**Author Contributions:** Conceptualization: S.H.B.; Methodology: S.H.B.; Formal Analysis: S.H.B. and M.B.; Investigation: S.H.B. and M.B.; Resources: S.H.B. and M.B.; Data Curation: S.H.B. and M.B.; Writing—original draft preparation: M.B.; Writing—review and editing: S.H.B. and M.B.; Visualization: S.H.B.; Supervision: S.H.B.; Project Administration: S.H.B. All authors have read and agreed to the published version of the manuscript.

**Funding:** This research received no external funding.

**Institutional Review Board Statement:** Not applicable.

**Informed Consent Statement:** Written informed consent has been obtained from the patient(s) to publish this paper.

**Data Availability Statement:** No additional data relating to this study is available.

**Conflicts of Interest:** The authors declare no conflict of interest.

**Appendix A**

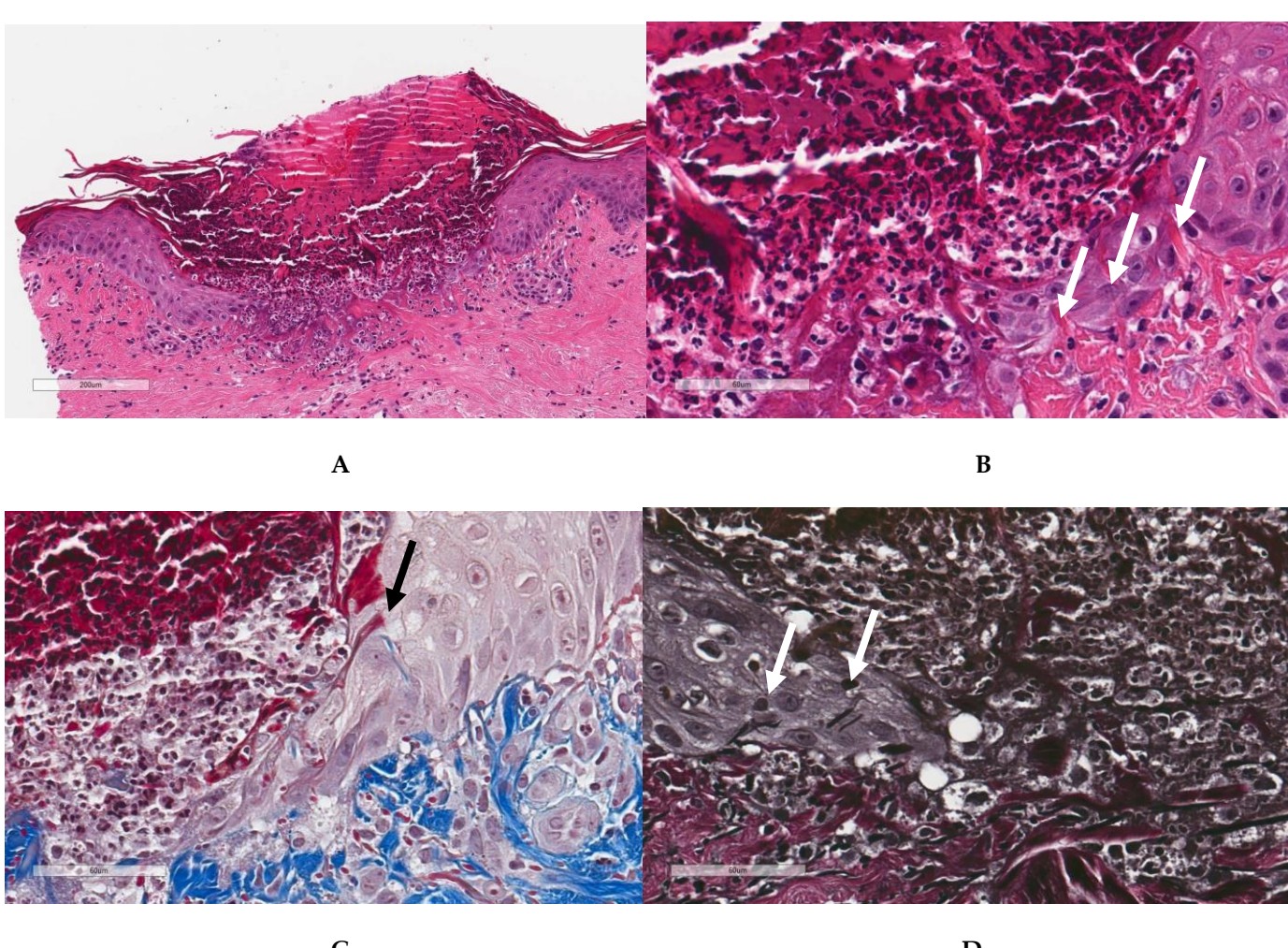

A

B

C

D

**Figure A1.** Reactive perforating collagenosis. (**A**) Hematoxylin and eosin-stained skin at low magnification shows a cup-shaped depression with degenerating collagen extending from the ulcer base into the crust. (**B**) High magnification showing the TE of collagen fibers between keratinocytes at one edge of the ulcer (white arrows). (**C**) Masson's trichrome confirms the presence of collagen fibers extruding between keratinocytes (black arrow). (**D**) Verhoeff elastic stain showing extrusion of elastic fragments between keratinocytes at one edge of the ulcer (white arrows).

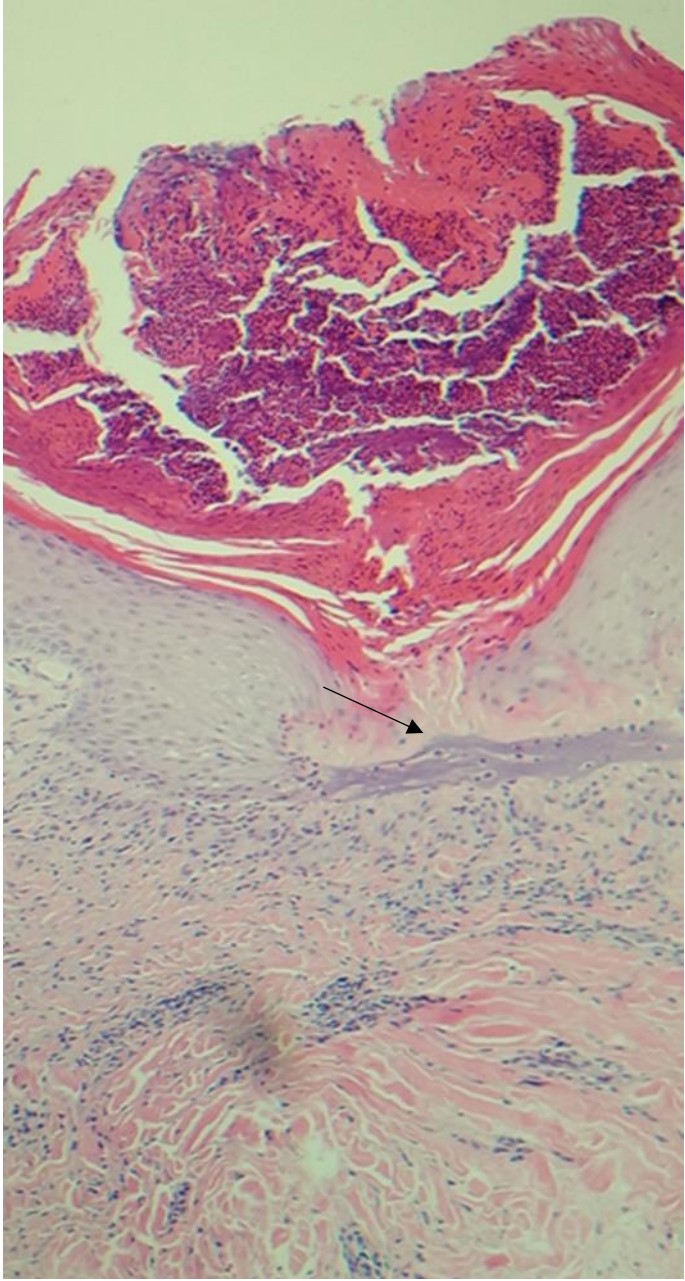

**Figure A2.** Perforating granuloma annulare. Hematoxylin and eosin-stained skin at 100× magnification shows hyperkeratosis, interstitial histiocytes, and TE of pale eosinophilic amorphous material consistent with mucin (black arrow).

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
