# Peer review of "Perforating Gout: Expanding the Differential for Transepidermal Elimination"

_dermatopathology, doi:10.3390/dermatopathology10030029_

Round 1

Reviewer 1 Report

1. The authors, after stating that Kyrle's disease is controversial, state in the table that the TE material is necrotic. What is the basis for this? If you can show this, it should be explained in the text.

2. While perhaps not a perforating disease, necrotizing crystalline folliculitis is a mimic for perforating gout, having a cup shaped surface (due to a distended follicular infundibulum) containing needle-like crystals.

Reviewer 2 Report

Thank you for your work on this topic, the paper provides a very comprehensive review of perforating dermatoses as well as a nice reminder to consider pseudocarcinomatous hyperplasia on transected shave biopsies. A few minor suggestions that might improve the paper:

- Line 91 - Please correct to "Kyrle's" instead of "Kryie's"

- Table 1 - For source instead of "Endogenous extracellular material or necrotic material" would put "Endogenous materials"

- Table 1 - For Kyrle's row would say "keratin and other materials" rather than necrotic material

- Table 1 - For Cells row instead of "Cells" would label as "Other" 

- Figure A2 - Would rotate photo so epidermis and dermis are vertically oriented. Since this is not a mucin stain (H&E) would be helpful to include an arrow highlighting the mucin

Thank you again for your work on this paper.

See above
